# Current Viewpoint on Female Urogenital Microbiome—The Cause or the Consequence?

**DOI:** 10.3390/microorganisms11051207

**Published:** 2023-05-04

**Authors:** Marina Čeprnja, Edin Hadžić, Damir Oros, Ena Melvan, Antonio Starcevic, Jurica Zucko

**Affiliations:** 1Biochemical Laboratory, Special Hospital Agram, Polyclinic Zagreb, 10000 Zagreb, Croatia; 2Department of Biochemical Engineering, Faculty of Food Technology and Biotechnology, Zagreb University, 10000 Zagreb, Croatia; 3Department of Biological Science, Faculty of Science, Macquarie University, Sydney, NSW 2109, Australia

**Keywords:** UT microbiota, NGS, cystitis, machine learning, bioinformatics, Big Data

## Abstract

An increasing amount of evidence implies that native microbiota is a constituent part of a healthy urinary tract (UT), making it an ecosystem on its own. What is still not clear is whether the origin of the urinary microbial community is the indirect consequence of the more abundant gut microbiota or a more distinct separation exists between these two systems. Another area of uncertainty is the existence of a link between the shifts in UT microbial composition and both the onset and persistence of cystitis symptoms. Cystitis is one of the most common reasons for antimicrobial drugs prescriptions in primary and secondary care and an important contributor to the problem of antimicrobial resistance. Despite this fact, we still have trouble distinguishing whether the primary cause of the majority of cystitis cases is a single pathogen overgrowth or a systemic disorder affecting the entire urinary microbiota. There is an increasing trend in studies monitoring changes and dynamics of UT microbiota, but this field of research is still in its infancy. Using NGS and bioinformatics, it is possible to obtain microbiota taxonomic profiles directly from urine samples, which can provide a window into microbial diversity (or the lack of) underlying each patient’s cystitis symptoms. However, while microbiota refers to the living collection of microorganisms, an interchangeably used term microbiome referring to the genetic material of the microbiota is more often used in conjunction with sequencing data. It is this vast amount of sequences, which are truly “Big Data”, that allow us to create models that describe interactions between different species contributing to an UT ecosystem, when coupled with machine-learning techniques. Although in a simplified predator—prey form these multi-species interaction models have the potential to further validate or disprove current beliefs; whether it is the presence or the absence of particular key players in a UT microbial ecosystem, the exact cause or consequence of the otherwise unknown etiology in the majority of cystitis cases. These insights might prove to be vital in our ongoing struggle against pathogen resistance and offer us new and promising clinical markers.

## 1. Introduction

The urogenital microbiota is the community of microorganisms that natively resides in the urinary and the genital tract. The notion of native microbial community residing in the urinary tract is fairly recent as even the Human Microbiome Project (HMP) sampled only gastrointestinal, oral, skin, nasal and genital (vaginal) sites. The urinary microbiota was not included within HMP in 2008 because urine was traditionally considered as being sterile in healthy, asymptomatic individuals. But, with the development of expanded quantitative urine culture (EQUC) and culture-independent techniques, such as genomic and proteomic ones, it was realized that the urinary system contains a diverse and complex community of microorganisms, which led to abandonment of previous clinical dogma. Current common belief is that healthy people have substantial microbiota in their urinary system, which differs between a healthy and diseased state as well as between the male and female sex [1,2,3,4,5,6].

The urinary microbiota is distinct from two closest niches inhabited by residual/native microbiota—gut and vagina—but still shares similarities with both of them. The current hypothesis on the origin of the urinary microbiota is that it may arise from several sources, including the gut microbiota, the vaginal microbiota (in females) and the environment. Studies have shown that the urinary microbiota is linked to the gut microbiota in terms of its composition and diversity, suggesting that some bacteria may migrate from the gut to the urinary tract [7]. In addition, research has also suggested that the vaginal microbiota may play a role in shaping the urinary microbiota in females, as certain bacteria found in the vagina can also be detected in the urine [6]. Finally, environmental factors such as exposure to bacteria in the external environment may also contribute to the composition of the urinary microbiota [8,9]. However, the exact mechanisms underlying the origin and maintenance of the urinary microbiota are still not fully understood and are an active area of research.

Healthy vaginal microbiota is specific from all other human body sites harboring native microbiota as its composition is, in the majority of the population, dominated by a single genus—*Lactobacillus*, which offers protection from pathogen colonization by lowering vaginal pH and impacting the host’s epithelial cells and immune response [10,11,12,13]. In attempts to classify vaginal microbiota types all being dominated by lactobacilli, four species have been identified as being distinct—*Lactobacillus iners*, *L. crispatus*, *L. gasseri* and *L. jensenii*—which appear in varying proportions as major parts of female microbiota worldwide [10,14,15,16]. Smaller proportion of females have a vaginal microbiota that is not dominated by lactobacilli, but is instead composed of a balanced population of obligate and facultative anaerobes, mostly characterized by genera *Gardnerella*, *Atopobium* and *Prevotella*. Due to the low presence of lactobacilli, this type of vaginal microbiota profile is associated with a higher risk of infections and adverse health outcomes [17,18,19]. The observed protective function of the vaginal microbiota characterized by low diversity and domination by *Lactobacillus* spp. is in contrast with the common belief linking diversity and “healthy” microbiota, but is supported by more than two decades of research into the vaginal microbiota.

Overall, the composition of healthy urinary microbiota as well as its role in maintaining our health and susceptibility to various diseases and ailments is just beginning to become elucidated [6,20]. The urinary tract is affected by interactions with nearby sites also hosting native microbiota, including the genital and digestive systems, which play a role in its dynamics [21]. So, it is of most importance to distinguish microorganisms native and residual to the UT from those transiently transferred from nearby body niches and their role in promoting health or disease. The definition of the “typical” makeup of the urinary microbiota and its possible protective or detrimental effects on the host is still an active area of research. Since urinary tract infections (UTIs) are among the most frequently diagnosed infections worldwide, which place a significant burden on healthcare systems and the lives of those affected, it would be beneficial to identify protective factors and to fully understand mechanisms microbiota plays in emergence of UTIs. With technological advancements, a greater understanding of microbial interactions and the introduction of machine learning and deep-learning methods into microbiome research, those efforts should soon bear fruit [22,23]. In this review, we will try to explore the urinary system from a perspective of a bioreactor inhabited by microbial communities and present an extensive, but not exhaustive list of optimal conditions for its steady state operation linked to the maintenance of health.

## 2. Urogenital Microbiome

The urinary microbiota and its more famous counterpart, the gut microbiota, are two distinct microbial communities that play important roles in human health [20,24,25]. While certain microorganisms are commonly found in the urinary microbiota, there is no specific microorganism that is unique to this environment, which has not been found in other bodily niches [1]. In fact, many of the microorganisms contributing to urinary microbiota are also found in other parts of the body, such as the gut, vagina or skin. For discussion about the urinary microbiota and its effect on the host, it is crucial to precisely identify microorganisms present in this low biomass niche. In order to accomplish this, the initial steps such as sample collection and sample preparation are of uttermost importance. Sample collection for the urinary microbiome can be troublesome as the most common and only noninvasive method, midstream urine sampling, can become contaminated from surrounding areas of the urethra, skin and genital tract; and thus, create unreliable/contaminated taxonomic profiles and lead to term genitourinary/urogenital microbiota for microbial communities obtained from voided urine samples in females [26,27,28,29].

It has to be stated that we are still missing the definition of commensal urinary microbiota regarding its composition, as is also the state with more researched microbial niches such as gut microbiota [30,31], and further complicated with the lack of proper definition of bladder health [26,32]. Current understanding on the composition of urogenital microbiota is based on a fairly limited number of publications with positive bias towards the female sex. Both male and female urinary microbiota are dominated by phylum *Firmicutes*, accounting from two-thirds to three-quarters of the overall phyla composition, followed by phyla *Actinobacteria*, *Bacteroidetes* and *Proteobacteria* [6,28]. Most commonly found genera in the urinary microbiota of healthy adults are members of *Lactobacillus*, *Gardnerella*, *Prevotella* and *Streptococcus*, as well as some members of *Escherichia*, *Enterococcus*, *Corynebacterium*, *Staphylococcus* and *Proteus* [27,28,33]. Attempts to define universal “urotypes” or “community structure types”, bacterial communities dominated by single taxa, have so far not resulted in consensus, but have confirmed previously identified genera as major contributors to the composition of urinary microbiota [34,35]. A study by Ugarcina Perovic et al. had identified much higher diversity at the species level within female urinary microbiota, indicating the underestimation of bacterial diversity by previous studies, while also highlighting discrepancies between methods used to characterize the bacterial composition of the female urinary microbiota [36]. It is worth noting that the composition of the urogenital microbiota varies between individuals and can be influenced by factors such as age, sex, diet and health status [8,9]. Changes in composition of endogenous urinary microbiota have been commonly associated with urinary tract infection, but it has also been correlated with kidney stones, incontinence, bladder cancer, prostate inflammation and cancer, among other things [29,37,38]. Since most of this insight comes from high-throughput DNA sequencing efforts, it is not actual microorganisms that are being observed, but instead it is their DNA fragments these conclusions have been drawn upon. Therefore, it is more accurate to use the term microbiome or, more specifically, “urobiome”, when referring to urinary microbiota. Urobiome is believed to play a role in the occurrence and prevalence of urinary tract infections (UTIs), which are one of most prevalent infections in the human population [39,40,41]. It is still unclear if disturbance in the urinary microbiota leading to dysbiosis is the underlying cause of UTIs or if it is the subsequent colonization of the urinary tract by bacteria from other bodily niches; or perhaps, these are simply different sides of the same coin [40,42]. So far, the research indicates that the dominant causative agent of UTIs, amounting to 80% of overall cases, can be attributed to uropathogenic *Escherichia coli* (UPEC) [6,43,44]. However, since it can also be present in commensal urinary flora and gut microbiota, its presence alone cannot be considered as a sole indicator of infection [45]. Currently, the “crosstalk” or interaction between bacteria seems to be the most persuasive explanation for the rise of pathogenicity of *E. coli* and other opportunistic members of the urinary microbiota [45,46,47]. These other members of gut microbiota that are often linked to the etiology of UTI are: *Klebsiella pneumoniae*, *Staphylococcus saprophyticus*, *Enterococcus faecalis* and *Proteus mirabilis*. For all these opportunistic pathogens, it was suggested their relative abundance in the gut is a risk factor for developing an UTI [40,48,49]. Since the gut is the most dominant microbiota niche, which appears to be tightly linked to UT, it is important to list the major differences between the two microbial niches, as was conducted in Table 1 below.

There are numerous virulence factors and mechanisms of interaction that can turn commensal bacteria in the urinary tract into pathogenic ones; the most prominent ones have been mentioned in Table 2.

It’s worth noting that different strains of pathogens may use different combinations of these factors, and the specific virulence factors involved can vary depending on the clinical presentation and severity of the infection.

## 3. Microbial Dynamics in Response to Change in Conditions

With time, all microbial communities change as a response to variations in environmental conditions or endogenous processes within microorganisms themselves [55]. Microbial communities adapt their endogenous dynamics by changing the community composition or by modulating gene expression within the population [56]. System dynamics gives crucial information about its function, but to fully understand microbiota stability and variability over time, more reliable data in the form of longitudinal studies are needed. Until now, only a few studies followed urogenital microbiota change through time and on a small number of study participants. Although few in numbers, these studies have shown higher interindividual variability of urinary microbiota, compared to gut microbiota, confirmed dynamics and resilience of urobiome and identified menstruation and vaginal intercourse as particular factors linked to observed changes [57,58].

Although the general awareness of the urobiome crucial role in human health is becoming better, its dynamic nature stays hidden. This is mainly because the established culture-dependent methods, such as bacterial culture and antibiotic sensitivity testing, which are predominantly being used in clinical settings, are not able to provide a “motion picture” view that is needed to capture the shifts in microbiome composition. In order to fully understand the diversity and functionality of this niche microbiome, it is paramount to start using culture-independent methods, such as 16S rRNA amplicon sequencing and whole-genome shotgun metagenomic sequencing, in addition to culture-dependent ones [59,60]. These methods enable the detection of bacteria that are difficult to culture using traditional microbiological techniques and they allow us to monitor the change in overall microbiome composition over time. By isolating collective DNA from urine samples using dedicated purification kits, variable 16S rRNA gene regions can be sequenced using the next-generation sequencing (NGS) platforms. Bioinformatics tools such as QIIME2 can then be used for downstream analysis, and taxonomic classification can be performed using mathematical methods such as the naive Bayes taxonomy classifier against the dedicated rRNA databases such as Greengenes or Silva [61]. The resulting amplicon sequence variants (ASV), which represent the variation observed in biological data that avoid clustering based on distance-based thresholds, are used as a proxy for taxonomic diversity [62]. Furthermore, if there are multiple such analyses being performed over a period of time on the same patient samples, it is even possible to model the dynamics of a microbial community linked to a health issue.

Mathematical models have been shown to be useful in the study of microbiota dynamics and several modeling methods, including network-based models that infer species interactions from abundance data [63,64], have been proposed in recent years to improve our understanding of these dynamics [65]. These interactions can be inferred using experimental time series data from microbial communities or cross-sectional co-occurrence data. The generalized Lotka-Volterra model (GLVM) is a common approach for modeling microbe–microbe interactions [66]. GLVMs are ordinary differential equation (ODE) systems that can be used to model logistic population growth, as well as pairwise species interactions, such as predator–prey dynamics, amensalism and competition. Predictive and dynamic modeling using first-order differential equations (e.g., GLVMs) is becoming more prevalent and it has provided valuable insights into microbial interactions and dynamics [67,68]. However, it requires massive datasets (as the number of parameters grows quadratically with the number of species) and dense longitudinal sampling, which limits the wider application of such methods [69]. With the lack of longitudinal study data due to limited research budgets, ethical concerns and the aforementioned challenges, generalized Lotka-Volterra models can also be fitted to cross-sectional data from multiple communities by imposing some additional assumptions [70,71]. Nevertheless, such methods still suffer from statistical and biological limitations. From a statistical perspective, the relative abundances of species inferred from microbiome sequencing data are not normally distributed and, furthermore, do not account for the measurement error when being extrapolated to actual microbiota abundances, contrary to the initial assumptions. From a biological standpoint: because the negative correlations are not accounted for, the proposed methods may erroneously infer competitive interactions among pairs of taxa where none exist. To address these issues, the GLVM parameters should be inferred from cross-sectional data using a Bayesian approach [72,73].

## 4. Factors Affecting Microbiota and Host Interactions

Urinary microbiota is a low biomass community located within the urinary tract where bacterial growth and colonization are affected by UT physiological parameters in combination with exogenous factors (Figure 1: for illustration purposes). In this section the physiological and exogenous factors that can affect urinary microbiota are being discussed and separated into host modifiable parameters and microbiota modifiable parameters.

### 4.1. Host Part of the Equation

Urine pH normally ranges from 4.5 to 8, tending to be slightly acidic as a result of metabolic activity [74]. Diets higher in fruit and vegetable, with lower meat intake were related to more alkaline urine, while diets rich in proteins and acidic fruits can cause more acidic urine [75]. Oxygen tension—oxygen availability is a crucial parameter for microbial growth and its availability plays a role in shaping the ecology of the urinary tract microbiota [20]. Recently, bladder urinary oxygen tension has been correlated with the composition of urinary microbiota and resultant clinical variables [76]. Specific gravity—urine-specific gravity is a measure of a person’s hydration status. It was shown that increased water intake in premenopausal women who drink low volumes of fluid daily can be an effective strategy to prevent recurrent cystitis [77]. Urine flow—the flow of urine through the urinary tract can help flush out non-attached or weakly adherent bacteria and prevent colonization [78]. Urine flow can be impacted by factors such as bladder function, urinary tract obstructions and catheterization. The flux of new urine is presumed to supply nutrients to resident microbes [79]. Urine composition—the composition of urine, including the presence of urea, electrolytes and other compounds, can impact bacterial growth and colonization by affecting the availability of nutrients and other factors that bacteria need to survive and thrive. Human urine is composed of many soluble elements, including electrolytes, osmolytes, amino acids and carbohydrates. The Urine Metabolome Database contains more than 2600 metabolites that have been detected in human urine [80]. Personalized analysis of urine composition may become part of a personalized combination therapy that includes antibiotics, antimetabolites and dietary interventions that affect urine composition [81]. Certain compounds present in food can be excreted in urine without being fully metabolized or absorbed by the body and these compounds can potentially affect the urinary microbiota. Here are some examples: Polyphenols—they belong to a group of plant compounds that are found in many foods, including fruits, vegetables, tea, coffee and red wine. Some polyphenols, such as cranberry proanthocyanidins, have been shown to have antimicrobial effects against bacteria in the urinary tract. Large variations of 34 dietary polyphenols in urinary excretion in the European population were largely determined by food preferences [82]. Fructooligosaccharides (FOS)—FOS are a type of prebiotic fiber that are found in many foods, including onions, garlic, bananas and artichokes. FOS are not fully digested by the body and can reach the colon intact, where they can stimulate the growth of beneficial bacteria. Common prebiotics are fructooligosaccharides (FOS), galacto-oligosaccharides (GOS), xylo-oligosaccharides (XOS) and lactulose [83]. A small fraction of FOC can be excreted in urine [84]. Cranberry-related compounds—Cranberries contain compounds such as proanthocyanidins and flavonoids, which have been shown to have antimicrobial effects against bacteria in the urinary tract. Cranberry juice has been shown to decrease the risk for UTI [85]. D-mannose—this is a type of sugar that is found in some fruits, such as cranberries, peaches and apples. D-mannose has been shown to have antimicrobial effects against E. coli, the commonest cause of uncomplicated UTIs. It is believed that presence of D-mannose in urine prevents the attachment of E. coli to the urothelium and, thus, prevents it from causing infection [86]. General consensus emerging in the last decade is that D-mannose, alone or in combination with other treatments, may be useful in the treatment of UTI/cystitis symptoms [87,88]. Probiotics—these live microorganisms have shown promise as possible novel therapeutics for the treatment and prevention of UTIs [89,90]. Although further investigations, especially clinical trials, are needed to fully and objectively assess their usefulness in the treatment of urological diseases [91], initial studies claiming the beneficial effect of several *Lactobacillus* and *Bifidobacterium* species are encouraging [91,92]. Still, more work is needed in answering questions of dosage, route of administration, length of therapy, selection of probiotic strains and their interaction with an individual’s urotype [93].

Finally, secretory immunoglobulins, such as secretory IgA (SIgA) and sex hormones, are also impacting urinary health. SIgA is an antibody whose main function is the immune exclusion at mucosal surfaces. This is a mechanism that prevents interaction of neutralized antigens with the epithelium. Another important function of SIgA is controlling the symbiotic relationship existing between commensals and the host, thus maintaining homeostatic regulation of urogenital mucosal epithelia [94,95]. Sex hormones—they were shown to play an important role in dynamics of urinary microbiota and microbiota in general [96], albeit in an indirect manner. Increased rate of rUTIs has been reported in postmenopausal women, along with the relative decrease in abundance of *Lactobacillus* species and increase in alpha diversity, which might be attributed to a decline in estrogen levels during and post menopause. Hormone replacement therapy was given for managing various urinary symptoms in women [97], but had been discontinued since discovery of its adverse effects [98,99]. While hormonal contraception use was associated with a significantly reduced risk of bacterial vaginosis [100], its impact on UTI is much less clear [101,102,103,104]. In addition, changes in microbiota dynamics were reported during the menstrual cycle [57].

In summary, dietary compounds, such as polyphenols, FOS, cranberry compounds, D-mannose and probiotics, show the potential to affect the urinary microbiota and confer favorable health outcomes for the host by modulating urinary microbiota. Host immune and endocrine systems also play an important role, but further investigations are needed to fully endorse their usage as alternative therapy for UT diseases.

### 4.2. Microbiota Part of the Equation

The urinary microbiota can produce a variety of molecules that may have effects on human health. Here are some examples of molecules that have been identified as secreted by the urinary microbiota, which may have effects on health: Short-chain fatty acids (SCFAs)—SCFAs are produced by some bacteria in the urinary tract and can have anti-inflammatory and immune-modulating effects. Studies have suggested that urinary SCFAs may be associated with a reduced risk of urinary tract infections and other urinary tract disorders. SCFAs play an important role in gut health. SCFAs play an important role in immune cell migration, cytokine production and the maintenance of cellular homeostasis. There is also increasing evidence that SCFAs play an important role in the gut–bladder axis and the gut–vagina–bladder axis crosstalk and affect the relationship between UTI and the intestinal microbiome [105]. SCFAs are mainly produced by two major clusters of bacteria, by Bacteroidetes (mainly propionate and acetate) and butyrate by Firmicutes [105,106]. Out of those, Firmicutes certainly play a major role in urinary microbiota, but Bacteroidetes are also a known constituent [6]. Nitric oxide (NO)—is known to be a part of the host antimicrobial defense [107]. NO can also be produced by some bacteria in the urinary tract either directly or indirectly through host stimulation, which can have anti-inflammatory and antimicrobial effects. Nitric oxide has been shown to play a role in regulating the immune response to urinary tract infections. Nitrites, which are usually not present in urine, are converted to nitric oxide (NO) and other reactive nitrogen oxides, which are toxic to a variety of microorganisms when acidified [108]. During the UTI, the major source of NO are inflammatory cells, especially neutrophils [109]. Some of the bacteria linked to increased NO production belong to species of the genus *Lactobacillus*, which are commonly found in the urogenital microbiota of healthy individuals. For example, *Lactobacillus crispatus* and *Lactobacillus jensenii* have been linked to increased NO production in vitro, and this production has been associated with their ability to inhibit the growth of pathogenic bacteria [110]. Quorum-sensing molecules—Some bacteria in the urinary tract use quorum-sensing (QS) molecules to communicate with each other. The aim of this approach is the creation of biofilm, which allows their survival in hostile environments. For example, it has been confirmed that QS is vital to virulence of a widespread urinary pathogen *Pseudomonas aeruginosa* [111]. These molecules may play a role in regulating bacterial colonization and the development of urinary tract infections. Gram-positive and Gram-negative bacteria use the same QS system mechanism to control bacterial gene expression; however, different signaling pathways are used in each group of bacteria [112]. Antibacterial peptides—the interplay of resident microbiota and host antimicrobial peptides are essential components of normal host innate immune responses that limit infection and pathogen-induced inflammation [113]. Epithelial cells lining the urinary tract prevent the adhesion of bacteria by producing a plethora of antibacterial peptides [113,114]. Resident microbiota, dominated by lactobacilli, can also produce molecules with antibacterial activities [115,116]. 

## 5. Bladder as a Bioreactor

The simplified function of the bladder can be compared to a vessel used to temporarily store urine in order to be voided and re-filled and, in this sense, it shares similarities to a bioreactor. As was mentioned in the previous section on physiological parameters, both the bladder and the urinary tract in general operate within precisely defined parameter limits. All those should be within optimal range for the urinary tract to function properly and maintain homeostasis. Bioreactor settings have similar requirements—appropriate temperature, nutrient supply and oxygenation are needed for the successful growth of cells or microorganisms. Both the bladder and a bioreactor can be affected by the presence of unwanted, harmful substances or contaminant microorganisms. In the bladder, the presence of pathogenic bacteria can cause infections and/or other urinary tract disorders, while in a bioreactor, contaminants can interfere with the growth and function of the material being processed. On the side of differences between the two, one must point out design, purpose and interaction with the environment (see Figure 2 for illustration purposes).

While the bladder is a natural organ and part of the human body, a bioreactor is a man-made device designed for specific purposes—including a simulation of the bladder [117,118]. The bladder is subject to biological processes and interactions, contained within the host organism and, as such, much harder to isolate from the surrounding tissues; on the other hand, although quite similar, a bioreactor is typically isolated from the external environment and designed to minimize interference from external factors. Concerning purpose, we must admit a bioreactor is more versatile regarding its application. A bioengineered bladder reactor, with the aim of being a fully functional human bladder substitute, is an ongoing process [119]. With the advances in technology and tissue engineering, it is conceivable that systems similar to ones designed for human gut microbiota [120,121] will also be designed and used to reveal dynamics of the residential urobiome, its interactions with vaginal and gut microbiota and its role in protection against UTIs and other urinary conditions. Urinary system microbiota, with its specifics such as low biomass and single species prevalence in its composition, poses specific challenges for in vitro systems, but successful efforts might give us insight into real time dynamics of the urobiome in relation to exogenous and endogenous conditions and offer better solutions for a healthy urinary system.

## 6. Food and Drugs Administration in Relation to Urinary Health

### 6.1. Probiotics and Urogenital Microbiota

Therapeutic advantages and benefits of probiotics are wide-ranging and depend on the strain [122,123,124]. Probiotic bacteria affect modification of the immune system, reduction of cholesterol, alleviation of lactose intolerance, alleviation or even remission of Crohn’s disease, prevention of diarrhea and prevention of urogenital infections [89,125,126,127]. The most common urotype at the genus level is *Lactobacillus*, followed by *Gardnerella*, *Corynebacterium*, *Streptococcus* and *Staphylococcus*, which are all Gram-positive bacteria, unlike the Gram-negative bacteria responsible for the majority of acute uncomplicated urinary tract infections [128]. Lactobacilli, as the dominant bacteria of the vaginal flora, have antimicrobial properties that regulate other urogenital microbiota, while changes in the distribution of bacteria of the urogenital microbiota can lead to infections [93]. Since lactobacilli dominate the urogenital flora of healthy women and have a protective role, it is suggested that the restoration of the urogenital microbiota by the application of probiotics, especially those lactobacilli-based, has the potential to prevent and stop urinary tract infections. Research over the last twenty years suggests that *Lacticaseibacillus rhamnosus* GR-1 and *Limosilactobacillus reuteri* RC-14 (previously called *Limosilactobacillus fermentum* RC-14) probiotic strains are showing potential for the prevention and treatment of UTIs. Moreover, studies have shown the effectiveness of *Lacticaseibacillus casei* Shirota and *L. crispatus* CTV-05 in a limited number of cases [129,130,131]. Further, some research suggests that the effectiveness of probiotics, such as *L. rhamnosus* GR-1 and *L. reuteri*, shows promise in the prevention or treatment of three main female urogenital infections: bacterial vaginosis, vulvovaginal candidiasis and urinary tract infections [90,132]. The oral administration of *Lacticaseibacillus*
*rhamnosus* GR-1 and *Limosilactobacillus fermentum* RC-14 for 60 days in one study [133] showed no harmful effects; moreover, microscopic analysis indicated a better restoration of the normal vaginal microbiota defined by the lactobacillus urotype, compared to the placebo group. Oral administration of the same probiotics in another research [134] showed similar results and the establishment of a healthy urogenital microbiota in up to 90% of study participants, but it also highlighted the importance of probiotic dosage. Furthermore, postmenopausal subjects having oral administration of *Lacticaseibacillus*
*rhamnosus* GR-1 and *Lacticaseibacillus*
*reuteri* RC-14 established a normal vaginal microbiota [135]. Probiotics therefore have a significant effect on the health of the urinary system. The combination of probiotics *L. rhamnosus* GR-1 and *L. fermentum* RC-14 is not only safe for daily usage in healthy women, but can significantly reduce the colonization of the vagina with potential pathogenic bacteria and fungi [133]. This is most likely due to lactobacili affecting pathogenic bacteria by producing lactic acid and, thus, lowering the pH of the environment, by adhering to the vaginal epithelium, so they can inhibit the attachment of pathogenic bacteria physically; and by competing for the same nutrients as pathogens, indirectly limiting their growth [136,137].

### 6.2. Prebiotics and Urogenital Microbiota

Research on the direct impact of prebiotics on the health of the urogenital tract is currently insufficient and the establishment of connection with healthy urogenital microbiota is possible only indirectly, through their effect on beneficial bacteria in the gut. Since the abundance of *Escherichia* and *Enterococcus* in the gut is a risk factor for the development of UTI [48], we can speculate that prebiotic supplementation with the aim of increasing lactobacilli abundance in the gut could have positive effect on the urinary tract [20]. The other, more direct, method of increasing abundance of lactobacilli in the gut would be a combination of synergic probiotic and prebiotic. *Limosilactobacillus reuteri* inhibits pathogens, such as *Salmonella typhimurium*, *Escherichia coli* and *Clostridium difficile*, by metabolizing glycerol into the antimicrobial compound 3-HPA or reuterin. The ability to excrete reuterin depends not only on the availability of glycerol, but also on glucose concentration [138]. An example of a synergic effect including prebiotic that stimulates growth of *Lacticaseibacillus rhamnosus*, which then produces the L-lactic acid isomer and ferments carbohydrates, would be: arabinose, cellobiose, esculin, ribose, sorbitol and sucrose [139]. Thus, prebiotics in this case are carbohydrates that cannot be digested by the host and are, therefore, available for enzymatic digestion by intestinal probiotics. It is known that two most studied prebiotics, inulin and galacto-oligosaccharides (GOS), stimulate the growth of a wide range of lactobacilli [140], but newer studies also indicate the need for prebiotics personalization [141]. Further research into the direct influence of prebiotics on the microbiome of the urinary tract is much needed.

### 6.3. Antibiotics and Urinary Tract Infections

In the treatment of urinary diseases, various antibiotics are being prescribed: Amoxicillin, Ceftriaxone, Cephalexin, Ciprofloxacin, Fosfomycin, Levofloxacin, Nitrofurantoin, Trimethoprim/sulfamethoxazole, to name a few. The duration of antibiotic therapy depends on the severity of the infection and varies from 3 to 14 days. More and more research is being sought on the advantages and disadvantages of shorter vs. longer therapy duration, and the complementary usage of probiotics as well as potential displacement of antibiotics with probiotics [142]. Some research [143] suggests that there is no significant difference between short-term (3–6 days) and long-term (7–14 days) periods of taking antibiotics. Vaginal infection is often characterized by a decrease in the number of native lactobacilli, and antimicrobial therapy is often ineffective at completely removing the symptoms of cystitis [144]. The assumption that the probiotics such as *Lacticaseibacillus rhamnosu*s GR-1 and *Limosilactobacillus reuteri* RC-14 can serve as an adjunct to antimicrobial treatment and improve the cure rate has been confirmed by research [145]. In other research, in addition to the antibiotic therapy, the participant groups took a probiotic and a placebo each. The experimental group taking probiotic combined with antibiotic had a cure rate of 87.5% compared to 50% for the group taking antibiotic and placebo. A study [146] showed that probiotic lactobacilli can also increase the effectiveness of the pharmaceutical agent against fungal pathogens in the treatment of urogenital diseases.

### 6.4. Nutrition and Healthy Urogenital Microbiota

While antibiotics are usually prescribed for treatment of UTIs, some foods can also help prevent or alleviate UTI symptoms. Cranberries and cranberry juice are examples of such foods. Studies have shown that cranberries and cranberry juice can help prevent and treat UTIs by preventing bacteria from adhering to the bladder walls [147]. Drinking 100% pure cranberry juice or taking cranberry supplements may be helpful. Probiotics can help improve gut and urinary tract health by increasing the number of beneficial bacteria in the gut. Foods such as yogurt, kefir, sauerkraut and kimchi are good sources of probiotics. Even simple measures such as drinking plenty of water can help flush out bacteria from the urinary tract and prevent UTIs. Certain spices such as garlic have antimicrobial properties that can help fight bacterial infections. Adding garlic to the diet or taking garlic supplements may be beneficial for preventing UTIs [148]. Moreover, vitamins such as vitamin C can help increase the acidity of urine, which can make it harder for bacteria to grow [149]. Foods rich in vitamin C include oranges, strawberries, kiwi fruit and bell peppers. It is important to note that while these foods may help prevent or alleviate UTI symptoms, they should not be used as a substitute for medical treatment. For the health of the urinary tract, the recommendations are to eat foods rich in nutrients, a diet that follows the pyramid of a healthy diet. Apart from cranberries and other fruits rich in vitamin C and other antioxidants [150], whole grains, sweet potatoes and similar foods rich in fiber supporting the control of insulin resistance [151], leafy vegetables for vaginal moisture support [152], with special emphasis on probiotics appear to be a general recommendation. The following are examples of such probiotics promoting food sources (especially *Lactobacillus*): yogurt with live cultures, sauerkraut and pickled vegetables, pickles, Kimchi (Korean national dish of pickled vegetables, sauerkraut), Miso (Japanese seasoning, fermented soybeans), Tempeh (traditional Indonesian food, fermented soy and starter culture) and Kombucha (Chinese tea created by yeast fermentation).

## 7. Conclusions

UTIs affect women more frequently than men. Cystitis is particularly prevalent in sexually active women and those who use certain types of contraception such as diaphragms or spermicidal agents. Other risk factors for cystitis include a weakened immune system, pregnancy, menopause and urinary tract abnormalities. Although it is generally not considered a serious health condition, it can cause discomfort and pain; and, in severe cases, it can lead to kidney infection or sepsis. In addition, recurrent episodes of infection can have a significant impact on a woman’s quality of life. Therefore, it is important for women to seek medical attention promptly. Treatment typically involves antibiotics, but lifestyle modifications, sometimes as simple as drinking more water and avoiding irritants such as caffeine or alcohol, can also help to prevent or manage cystitis. Recent research has shown that the urinary microbiota may play an important role in the development of cystitis. Studies have found that alterations in the composition of the urogenital microbiota, such as an overgrowth of pathogenic bacteria or a reduction in beneficial native bacteria, can contribute to the development of cystitis. This may occur due to a breakdown in the protective mechanisms that prevent the ascent of bacteria from the urethra into the bladder. In addition, research has shown that certain species of bacteria such as *E. coli* are commonly associated with cystitis. *E. coli* is a normal inhabitant of the gut microbiota, but can cause infection if it colonizes the urinary tract. Understanding the role of the urogenital microbiota in the development of UTIs is an active area of research, and it may lead to the development of new strategies for preventing and treating this common condition. Modern high-throughput next-generation sequencing methods provide us with much-needed metagenomic datasets that can be used to model actual microbial community dynamics within every bodily niche, especially UT since it is simpler than other niches such as gut or skin. This novel approach involves analyzing the interactions between different species of microorganisms, such as the competition for resources or cooperation in the production of metabolites, and how these interactions affect the overall composition of the microbiota. Based on this insight, researchers can gain a better understanding of how changes in the composition of the urogenital microbiota contribute to the development of UTIs. The new predominantly culture-independent analyses based on high-throughput technologies have shed new light on the complex and dynamic nature of the urogenital microbiome, highlighting the importance of the microbiome in UTI development and management. The urogenital microbiome is a diverse and complex ecosystem, comprising a variety of microorganisms, including bacteria, fungi, viruses and protozoa. The microbiome plays an essential role in maintaining the health and function of the urinary tract by providing colonization resistance against potential pathogens, modulating the host immune response and contributing to metabolic and biochemical processes. Dysbiosis of the urogenital microbiome, characterized by alterations in microbial composition and diversity, has been associated with increased susceptibility to UTIs, recurrent infections and antibiotic resistance. Despite the growing evidence supporting the role of the urogenital microbiome in UTI pathogenesis, many knowledge gaps still need to be addressed to fully understand the complexity of this ecosystem. For example, the mechanisms underlying the interplay between the microbiome and the host immune response, the impact of lifestyle factors on microbiome composition and function and the potential of novel microbiome-based therapies for UTI prevention and treatment remain poorly understood. Future research in the field should aim to address these gaps by combining high-throughput sequencing and metagenomic analysis with functional studies, to better understand the molecular mechanisms underlying UTI pathogenesis and to identify potential targets for therapeutic intervention. Additionally, more studies are needed to elucidate the role of fungal and viral components of the urogenital microbiome in UTI development and management, which are being neglected. It is only with greater understanding of the host–microbiota relationship and with more accessible new research techniques, a more personalized approach to therapy, including a more savvy usage of therapeutics and the synergistic effects of food, that probiotics and prebiotics will be able to pave a road to the more sustainable management of UTIs.

## Figures and Tables

**Figure 1 microorganisms-11-01207-f001:**
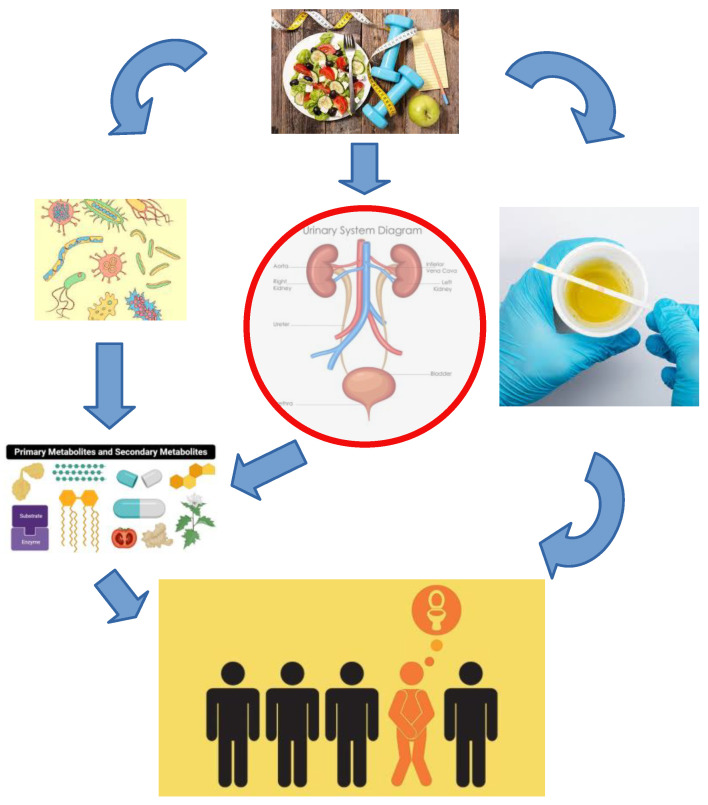
An illustration of interconnected relations between microbiota and host mediated through host modifiable parameters such as diet, exercise, hormones, habits and microbiota modifiable parameters such as excreted metabolites and virulence factors, which make a difference between healthy and diseased urotype.

**Figure 2 microorganisms-11-01207-f002:**
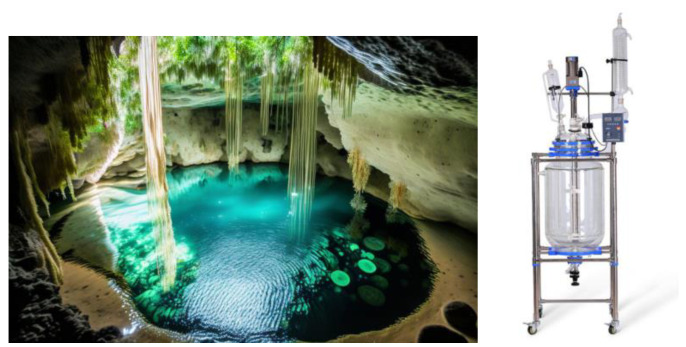
An AI-made illustration of a natural phenomenon—a Cenote, surface connection to subterranean water bodies, usually represented by small sheltered sites with no surface exposed water containing small numbers of bacteria, next to a stirred jacket glass bioreactor. The Cenote illustration was made by Deep Dream Generator, https://deepdreamgenerator.com (accessed on 25 March 2023).

**Table 1 microorganisms-11-01207-t001:** Key differences between urogenital and gut microbiota.

Feature	Characteristics
Composition	The urobiome has been observed in several urotypes, usually dominated by a single species or genus. Most common are urotypes dominated by *Lactobacillus*, but identified were also ones dominated by bacteria such as *Gardnerella*, *Prevotella* and *Streptococcus* or without dominant taxa. The urogenital microbiome was also described in several subtypes following similar distribution and composition as the urobiome. The gut microbiome is significantly more complex, without a single species dominating.
Location	The urogenital microbiome is found in the female reproductive tract and in the male and female urinary tracts. The gut microbiome, on the other hand, is found in the gastrointestinal tract.
Function	The urogenital microbiome plays a critical role in maintaining a healthy vaginal pH and preventing infections, such as urinary tract infections and bacterial vaginosis. The gut microbiome, in contrast, is involved in digestion and absorption of nutrients, as well as immune system regulation.
Diversity	Urobiome is less diverse than its intestinal counterpart. It is usually dominated by a single species or genus. The gut microbiome is much more diverse, with hundreds of different species of microorganisms living in the gut. This diversity is important for maintaining overall health, as disruptions to the gut microbiome have been linked to a variety of health conditions, such as inflammatory bowel disease and obesity.

**Table 2 microorganisms-11-01207-t002:** Five most common examples of virulence factors and/or mechanisms of interaction that can turn otherwise harmless urinary tract bacteria into potential pathogenic ones.

Factor	Characteristics
Adhesion and colonization	Pathogenic bacteria can adhere to and colonize the urinary tract by using surface structures called pili and fimbriae to bind to specific receptors on the epithelial cells that line the urinary tract [50]. This allows the bacteria to establish a foothold in the urinary tract and avoid being flushed out by urine flow.
Biofilm formation	Once pathogenic bacteria have colonized the urinary tract, they can form biofilms, which are dense aggregates of bacteria encased in a matrix of extracellular polymeric substances [51]. Biofilms can protect the bacteria from the host immune system and antimicrobial agents, making them more difficult to eradicate.
Production of toxins	Pathogenic bacteria can produce a range of toxins that can cause damage to host cells and tissues. For example, some strains of E. coli produce hemolysins, which can cause lysis of red blood cells and damage to epithelial cells in the urinary tract [52].
Type III secretion system	Pathogenic bacteria can also use a type III secretion system (T3SS) to inject effector proteins into host cells, altering their function and contributing to pathogenesis. The T3SS is a specialized protein secretion system that allows the bacteria to directly manipulate host cell signaling pathways, leading to changes in cell morphology, cytokine production and other immune responses [53].
Antibiotic resistance	Some pathogenic bacteria have acquired resistance to multiple antibiotics, making them more difficult to treat and increasing the risk of recurrent infections [54].

## Data Availability

No new data were created or analyzed in this study. Data sharing is not applicable to this article.

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
