# Peer review of "Current Viewpoint on Female Urogenital Microbiome—The Cause or the Consequence?"

_microorganisms, 2023, doi:10.3390/microorganisms11051207_

Round 1
Reviewer 1 Report
As a general comment that applies to most sections, do not confuse the terms urinary and urogenital microbiota. Urinary microbiota refers to the urinary microbial community studied using urine samples obtained by transurethral catheterization or suprapubic puncture. Urogenital microbiota refers to the urinary microbial community studied by midstream urine samples with genital contamination. Please check this along the text.
Bacterial genera and species should be in italics. Please check these terms along the text.
Section 1. Introduction.
Second paragraph. ‘Normal vaginal microbiota’ should be replaced by ‘Healthy vaginal microbiota’.
Third paragraph. The term urogenital refers to the combination of the urinary tract and genitalia. Thus, ‘urogenital and digestive systems’ should be replaced by ‘genital and digestive systems’.
As indicated in the text, the urinary microbiota is strongly influenced by the intestinal and genital microbiota. In fact, there are some hypotheses that propose either of these two niches as the origin of the urinary microbiota. In this sense, it would be interesting to include in the text information on the origin of the urinary microbiota.
Section 2. Urogenital microbiome.
Second paragraph. “Normal urinary microbiota” should be replaced by “commensal urinary microbiota”.
It would be interesting to explain the virulence factors or mechanisms of interaction with other bacteria that turn a commensal E. coli into pathogenic.
The term “bacteriuria” can lead to confusion. The presence of a commensal urinary microbiota has already been recognized, so the use of the term asymptomatic bacteriuria to refer to the presence of bacteria in urine would not be correct.
Table 1 shows that the urogenital microbiome is mainly composed of Lactobacillus while the urobiome presents several urotypes. This is not very accurate; the urogenital microbiome, like the urinary microbiome, can present different urotypes, one of them being dominated by Lactobacillus although it is not the only one. In the location section of table 1, the intestinal microbiome location is not present; please check this. In addition, this table shows that “Urobiome is not diverse. It is usually dominated by a single species or genus”. The presence of urotypes and, therefore, microbial communities dominated by one or two bacterial genera does not indicate a lack of diversity. They are less diverse communities than those present in the intestinal microbiome, but they are still not lacking in diversity.
Section 4. Factors affecting microbiota and host interactions
This section describes several factors that interact with the urinary microbial community. A figure showing the interactions between the host and bacterial factors and the urinary microbiota would greatly enhance the article.
Author Response
Dear Reviewer,
We found your comments very constructive and exact and hopefully we have managed to answer them all in our revised version of the manuscript. All the changes were made under "track records" option, so you have a clear outlook on what has been done with the manuscript.
To summarize, we have followed your suggestions regarding the "general" terminology along the entire manuscript. This goes for the terms urinary and urogenital microbiota as well as "normal" vs "healthy". All your suggestions have been accepted and text has been changed accordingly.
In the introduction, we have included information on the origin of the urinary microbiota, and we have tried to explain the virulence factors or mechanisms of interaction with other bacteria that turn a commensal bacteria into pathogenic ones.
We also found your suggestion about introducing a new figure summarizing factors affecting microbiota and host interactions as a welcome addition so we have done accordingly.
Once again, thank you for dedicating your time in order to improve our effort.
Kind regards!
Reviewer 2 Report
The manuscript: Current viewpoint on Female Urogenital Microbiome - the cause or the consequence?
Marina Čeprnja, Edin Hadžić, Damir Oros, Ena Melvan, Antonio Starcevic, Jurica Zucko
Title: The title does not correspond to what is written in the text of the article. The name says - urogenital microbiome, and the text speaks mainly about the microbiome of the bladder. Sometimes there is a mention of the gut microbiome.
Introduction: enough; represent the essence of the problem; does not require change
Methodology: meets the requirements of the journal and the branch of knowledge; does not require modification;
Results: Section 2. has the name «Urogenital Microbiome». But in the text, all attention is paid to the microbiome of the bladder. Although table 1 also shows the characteristics of the urogenital microbiome.
Section 3 - Microbiome dynamics. Modeling to study the dynamics of the microbiome is very useful, but this section does not provide specific data on the variability of the microbiome of different loci. And what significance do these changes have for practice
Section 4. Factors affecting microbiota and host interactions is also devoted to urine studies and the microbiome of the urinary tract
Section 5. Blade as a bioreactor - also devoted to the study of the microbiota of the bladder
And finally, section 6. Food and drug administration in relation to urinary health - bladder again
Moreover, the microbiome of the bladder in women and men is compared
Literature: sufficient for the article and does not require additions
Article design: meets the requirements of the journal
Conclusion: The conclusion is also devoted to studies of the bladder in both women and men
My suggestion is to change the name: The modern point of view on the female urogenital microbiome - cause or effect? - on: A modern point of view on the microbiome of the bladder
Author Response
Dear Reviewer,
Thank you for your comments, we have incorporated both yours and another reviewer's comments into a revised version of the manuscript and we hope that it will be more agreeable in respect to your comments.
As we understood, the major issue was the "bladder" vs more "general" urinary (or urogenital) microbiome. Our intention was not to overemphasize bladder over the general UT system. The urinary tract is a complex system that includes the kidneys, ureter, bladder, and urethra, and each of these organs plays a critical role. It's the same when it comes to UT microbiota. The microbiota of the urethra and other parts of the urinary tract, all probably play important roles in maintaining urinary health and preventing infections. That being said, we have based this manuscript on NGS data viewpoint obtained mostly by performing 16S rRNA amplicon sequencing of urine samples.
In this regard, the urine being analyzed has arguably spent most of it's time inside bladder, before being voided and collected. The resulting microbiome most certainly reflects the entire UT system, however if we follow the analogy with our gut microbiota (on which more is known), in which the colon is perceived as most important simply based on the overall microbial abundance, same thing could be argued for the urethra and bladder.
Hope that this clarification explains direction of our manuscript. We simply wanted to highlight the possibilities of using urine and NGS in order to obtain insight into urinary microbiome and use this data to both optimize existing treatments or perhaps suggest dietary changes, which could lead to more healthy UT. Regarding the role of modelling in this perspective, we have actually done some work in this direction (please take a look at: Ceprnja M, Oros D, Melvan E, Svetlicic E, Skrlin J, Barisic K, Starcevic L, Zucko J, Starcevic A. Modeling of Urinary Microbiota Associated With Cystitis. Front Cell Infect Microbiol. 2021 Mar 16;11:643638. doi: 10.3389/fcimb.2021.643638. PMID: 33796485; PMCID: PMC8008076.).
Regarding the title, hopefully after reading the revised version of the manuscript where we worked on a more unison terminology and added some explanatory paragraphs, it will sound better, but if not, please let us know because we are willing to make alternations, after consulting with the Editor.
With regards!